# Ectodermal Dysplasia-Syndactyly Syndrome with Toe-Only Minimal Syndactyly Due to a Novel Mutation in NECTIN4: A Case Report and Literature Review

**DOI:** 10.3390/genes12050748

**Published:** 2021-05-17

**Authors:** Roberta Rotunno, Andrea Diociaiuti, Maria Lisa Dentici, Martina Rinelli, Michele Callea, Chiara Retrosi, Giovanna Zambruno, Emanuele Bellacchio, May El Hachem

**Affiliations:** 1Dermatology Unit and Genodermatosis Unit, Genetics and Rare Diseases Research Division, Bambino Gesù Children’s Hospital, IRCCS, Piazza Sant’Onofrio 4, 00165 Rome, Italy; roberta.rotunno@opbg.net (R.R.); chiara.retrosi@opbg.net (C.R.); may.elhachem@opbg.net (M.E.H.); 2Medical Genetics Unit, Bambino Gesù Children Hospital, IRCCS, Piazza Sant’Onofrio 4, 00165 Rome, Italy; marialisa.dentici@opbg.net; 3Laboratory of Medical Genetics, Translational Cytogenomics Research Unit, Bambino Gesù Children’s Hospital, IRCCS, Piazza Sant’Onofrio 4, 00165 Rome, Italy; martina.rinelli@opbg.net; 4Dentistry Unit, Bambino Gesù Children Hospital, IRCCS, Piazza Sant’Onofrio 4, 00165 Rome, Italy; michele.callea@opbg.net; 5Genodermatosis Unit, Genetics and Rare Diseases Research Division, Bambino Gesù Children’s Hospital, IRCCS, Piazza Sant’Onofrio 4, 00165 Rome, Italy; giovanna.zambruno@opbg.net; 6Molecular Genetics and Functional Genomics Unit, Genetics and Rare Diseases Research Division, Bambino Gesù Children’s Hospital, IRCCS, Piazza Sant’Onofrio 4, 00165 Rome, Italy; emanuele.bellacchio@opbg.net

**Keywords:** ectodermal dysplasia, syndactyly, nectin-4, hypotrichosis, hypodontia, enamel hypoplasia, palmoplantar keratoderma, trichoscopy

## Abstract

Ectodermal dysplasia-syndactyly syndrome 1 (EDSS1) is characterized by cutaneous syndactyly of the toes and fingers and abnormalities of the hair and teeth, variably associated with nail dystrophy and palmoplantar keratoderma (PPK). EDSS1 is caused by biallelic mutations in the NECTIN4 gene, encoding the adherens junction component nectin-4. Nine EDSS1 cases have been described to date. We report a 5.5-year-old female child affected with EDSS1 due to the novel homozygous frameshift mutation c.1150delC (p.Gln384ArgfsTer7) in the NECTIN4 gene. The patient presents brittle scalp hair, sparse eyebrows and eyelashes, widely spaced conical teeth and dental agenesis, as well as toenail dystrophy and mild PPK. She has minimal proximal syndactyly limited to toes 2–3, which makes the phenotype of our patient peculiar as the overt involvement of both fingers and toes is typical of EDSS1. All previously described mutations are located in the nectin-4 extracellular portion, whereas p.Gln384ArgfsTer7 occurs within the cytoplasmic domain of the protein. This mutation is predicted to affect the interaction with afadin, suggesting that impaired afadin activation is sufficient to determine EDSS1. Our case, which represents the first report of a NECTIN4 mutation with toe-only minimal syndactyly, expands the phenotypic and molecular spectrum of EDSS1.

## 1. Introduction

Ectodermal dysplasias (EDs) comprise a broad and heterogeneous group of genetic disorders affecting the development and/or homeostasis of two or more ectodermal derivatives, including the hair, teeth, nails, eccrine and other glands [1,2]. More than 200 clinically distinct isolated and syndromic forms of EDs have been described, with an estimated cumulative incidence rate of 7 in 10,000 births [1,2,3]. Progress in deciphering the molecular basis and pathogenesis of an ever-growing number of EDs has led to the proposal of a new classification system that integrates both clinical and molecular information [1]. ED-syndactyly syndrome 1 (EDSS1; OMIM#613573) is a rare type of ED caused by biallelic mutations in the NECTIN4 gene (previously known as poliovirus-like receptor 4, PVRL4) located on chromosome 1 (1q23.1–3) [4]. NECTIN4 encodes the cell adhesion molecule nectin-4, highly expressed in adherens junctions of the suprabasal epidermis and in the hair follicle [4]. EDSS1 is characterized by sparse to absent scalp hair, eyebrows and eyelashes, abnormal dentition (peg-shaped and conical crowns and enamel defects), hypoplastic nails, palmoplantar keratoderma and bilateral partial cutaneous syndactyly, variably affecting the fingers and toes [4,5]. Few NECTIN4 mutations have thus far been described in EDSS1 and no genotype-phenotype correlations have yet been reported. However, three clinical features (cutaneous syndactyly and abnormalities of the hair and teeth) are reported as constant in EDSS1 despite the different NECTIN4 causative mutations [4,5,6]. Here, we report on a 5.5-year-old girl who is suspected of being affected by ED based on the presence of hair and tooth abnormalities and toe-nail dystrophy. Syndactyly is minimal and limited to toes 2–3. Molecular genetic testing discloses a novel homozygous truncating variant of the NECTIN4 gene. Our findings expand the phenotypic and molecular spectrum of this rare condition.

## 2. Materials and Methods

After gaining informed consent, the patient and parents’ genomic DNA was extracted from peripheral blood using a QIAsymphony DSP DNA Mini Kit (Qiagen, Hilden, Germany). Molecular characterization of the patient was performed by next-generation sequencing (NGS), using a custom-panel kit (NimbleGen SeqCap Target Enrichment, Roche, Madison, WI, USA, on NovaSeq 6000 system), including ED-associated genes. The BaseSpace pipeline [7] and TGex software (LifeMap Sciences, Inc. 4.0, Alameda, CA, USA) were used for identifying and annotating variants, respectively. Identified variants were evaluated by VarSome [8] and classified according to the American College of Medical Genetics and Genomics (ACMG) guidelines [9].

## 3. Results

### 3.1. Case Report

A 5.5-year-old girl was referred to our Rare Skin Disease Center for suspected ED. She was born at term by cesarean section for a fetal malposition. The parents were healthy with an unremarkable family history; they were apparently unrelated but originated from a small village (about 1000 inhabitants). The proband attained normal developmental milestones except for a language delay, then resolved. At five years of age, she underwent percutaneous closure of an ostium secundum atrial septal defect without complications.

Physical examination revealed mild hypotrichosis with brittle scalp hair, which was uncombable, slow-growing and never required cutting (Figure 1A). Her eyebrows and eyelashes were sparse (Figure 1A). At trichoscopy, the hair showed morphological abnormalities including twists at irregular intervals (pili torti-like) (Figure 2A). Her teeth were widely spaced and conical, with small crowns and enamel hypoplasia (Figure 1B); agenesis of four permanent teeth (the four wisdom teeth) was documented by orthopantomogram (Figure 2B). The patient had minimal and hardly noticeable proximal cutaneous syndactyly, limited to toes 2–3, and toenail dystrophy (Figure 1C). Additionally, she had diffuse xerosis, keratosis pilaris of the cheeks, arms, thighs and buttocks, and mild palmoplantar hyperkeratosis (Figure 1D). Sweating was normal. Her ears and eyes were morphologically and functionally normal. 

### 3.2. Molecular Genetic Analysis

Due to the phenotypic presentation, we confirmed the suspected diagnosis of ED. NGS analysis revealed a previously undescribed homozygous frameshift variant in exon 6 of the NECTIN4 (NM_030916) gene, c.1150delC, p.Gln384ArgfsTer7 (Figure 3A). Sanger sequencing confirmed the homozygous and heterozygous status of the variant in the proband and her parents, respectively (Figure 3B). The variant was classified as pathogenic according to the ACMG guidelines [9] and the consistent clinical phenotype. 

## 4. Discussion

Defective cell–cell adhesion may result in different EDs [10,11]. In particular, mutations in NECTIN1 and NECTIN4, the genes encoding nectin-1 and -4, cause cleft lip/palate ED (CLPED1; OMIM#225060) and EDSS1, respectively [4,10,11,12]. Hair and tooth abnormalities and cutaneous syndactyly are clinical features common to both EDSS1 and CLPED1, the latter being typified by the additional presence of a cleft lip/palate. In 2013, Brancati et al. proposed grouping these entities under the term “nectinopathies” [5]. Four distinct nectins are known. They are calcium-independent immunoglobulin-like (Ig-like) transmembrane cell adhesion molecules (CAMs) working in cell–cell junctions, especially at adherens junctions (AJ) in cooperation with cadherins [10]. The 55.5-kDa adhesion receptor nectin-4 consists of 510 amino acids and is encoded by the poliovirus receptor-related 4 (NECTIN4/PVRL4) gene [10]. Nectin-4 is a multidomain protein formed by an N-terminal extracellular domain containing three immunoglobulin-like subdomains (one V-type and two C2-type), a transmembrane domain and a cytoplasmic domain, which is connected to the actin cytoskeleton through afadin, an F-actin-binding protein [10]. Homophilic and heterophilic interactions occur through Ig-like domains of nectins. In particular, nectin-1 and -4 trans-interactions promote and regulate different developmental pathways like actin cytoskeleton reforming through the activity of a member of the Rho family of small GTPases, Rac1, which further promotes adhesion molecule clustering and strengthens their connections [13,14]. Loss/impairment of nectin-4 leads to loss/impairment of Rac1 activation by defective transdimerization of nectin-1 and nectin-4 [14]. These data outline a synergistic action of nectin-1 and -4 in the early steps of AJ formation and implicate this interaction in modulating the Rac1 signaling pathway. The nectin-4 molecule is strongly expressed in the epidermis, hair follicle structures and cultured keratinocytes, as well as in the embryo and placenta [14]. Brancati et al. suggested its role in hair morphogenesis and cycling. Mutant nectin-4 leads to defective organogenesis [4]. Cutaneous syndactyly results from cell apoptosis in the interdigital tissue involving different signaling pathways, and nectin-4 is specifically expressed in the last phases of digit separation in the mouse embryo [4,15].

EDSS1 is a very rare autosomal recessive disorder. We report a novel C-terminal homozygous frameshift mutation (p.Gln384ArgfsTer7) in NECTIN4 in a 5.5-year-old female, affected by EDSS1. Her clinical presentation was unique in that she presented with minimal cutaneous syndactyly. A literature review identified three sporadic EDSS1 cases and six families with two or more affected members (Table 1) [4,6,14,16,17,18,19,20]. The patients originated from Denmark, Turkey, Algeria, Italy, Afghanistan, Pakistan, Azad Jammu and Kashmir. Consanguinity was reported in seven families [4,6,14,17,18,19,20]. A proximal bilateral cutaneous syndactyly of both fingers and toes was present in previous cases [4,14,17,18,19,20], with the exception of a girl who had involvement limited to toes [6]. In our patient, cutaneous syndactyly was barely visible and affected only toes 2–3. Our findings underlie the need for careful physical examination in order to identify this typical sign of EDSS1. Our patient had mild scalp hypotrichosis with fragile hair and sparse eyelashes and eyebrows. Hypotrichosis is a constant and early finding in EDSS1; it has a progressive course, which can lead to alopecia in adulthood [4,6,14,16,17,19,20]. In addition, a family with congenital alopecia has been described [18]. Similar to previous reports, our child had widely spaced teeth with conical-shaped crowns and enamel defects [4,6,16,17,18,19,20]. Moreover, orthopantomogram showed hypodontia, which has been previously described in a single family [18]. However, this finding might be underestimated as radiographic evaluation is not reported in other studies and some patients were children with deciduous dentition only [6,19]. Our patient also presented mild toenail dystrophy and palmoplantar hyperkeratosis, features variably present in EDSS1 [17,18,20]. Similar to most previous cases, she reported normal sweating [4,6,16,17,19]. However, a palmar biopsy revealed the complete absence of sweat gland units in a single family complaining of heat intolerance [18]. Additional clinical findings reported in EDSS1 patients include: bilateral purulent conjunctivitis in a single family, ear deformities in two families and slightly hypoplastic nipples in a sporadic case [18,19,20].

To date, only ten NECTIN4 mutations have been described in nine families with EDSS1; all the mutations were private and no genotype-phenotype correlations have yet been formulated [4,6,14,16,17,18,19,20]. Pathogenic sequence variants comprised two nonsense and a frameshift mutation as well as six missense (Table 1 and Figure 4) [4,6,14,17,18,19,20]. In silico analysis combined with previous functional studies supports a deleterious effect on gene products of all missense mutations described. The previously reported missense and nonsense/frameshift mutations of nectin-4 hit the ectodomain of the protein-producing ectodermal dysplasia and cutaneous syndactyly of the hands and feet. The Gln384ArgfsTer7 mutation identified in our patient occurs more C-terminally, in the cytoplasmic portion of the protein (Figure 4). Although still causing the modification/loss of a long amino acid region (127 residues), the phenotype for this mutation appears milder, in particular regarding syndactyly. The mutated region shares little conservation among nectins 1–4 with the exception of the most C-terminal residues (Figure 4) known to mediate the interaction with the PDZ domain of afadin [21]. Since all mutant NECTIN4 identified so far in EDSS1 carry a truncation or a structural defect in the N-terminal extracellular domain, mutations affecting the ectodomain, which mediates homophilic and heterophilic nectin interactions, might produce a full phenotypic presentation of EDSS1. On the other hand, milder phenotypes would result from truncating mutations that only affect the less conserved cytoplasmic tail. Nevertheless, mutation Gln384ArgfsTer7 that causes the loss of interaction of the nectin carboxyl terminus with afadin is sufficient for the manifestation of the EDSS1 phenotype. In conclusion, we have identified a novel homozygous variant in the NECTIN4 gene causing EDSS1 with minimal cutaneous syndactyly limited to the feet in an Italian girl. Our data show that mutations in the NECTIN4 gene may fall in the C-terminal portion of the gene causing a mild EDSS1 phenotype, and further contribute to defining the disease’s clinical spectrum.

## Figures and Tables

**Figure 1 genes-12-00748-f001:**
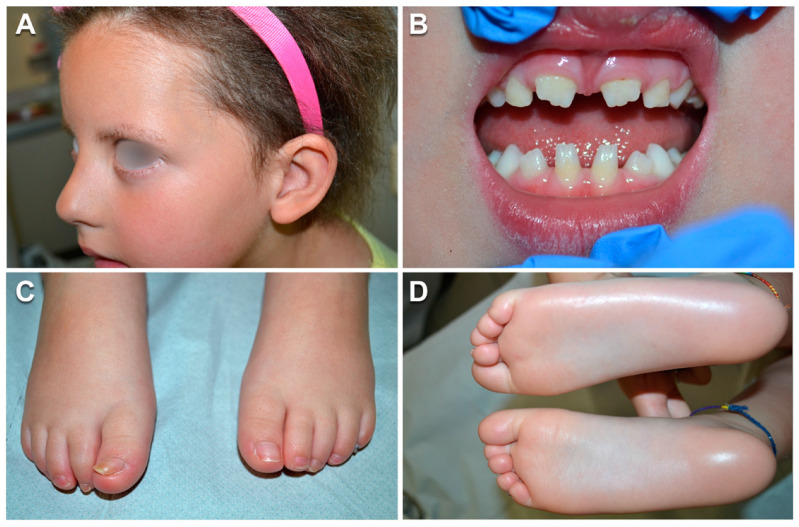
Patient’s clinical features. Mild hypotrichosis with brittle scalp hair and sparse eyebrows and eyelashes at age 5.5 years (**A**), widely spaced teeth with conical, small crowns and enamel hypoplasia (**B**), minimal proximal syndactyly limited to toes 2–3 and onychodystrophy (**C**) and mild plantar hyperkeratosis (**D**).

**Figure 2 genes-12-00748-f002:**
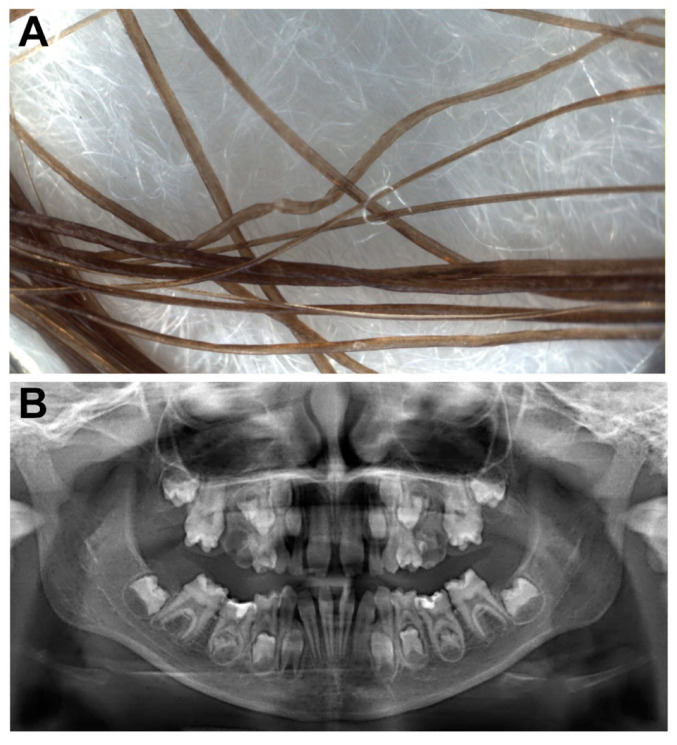
Instrumental examinations. Trichoscopy showing hair twisted at irregular intervals (pili torti-like) (**A**) and an orthopantomogram documenting agenesis of the four wisdom teeth, small crowns and diffuse enamel hypoplasia of several permanent teeth (**B**).

**Figure 3 genes-12-00748-f003:**
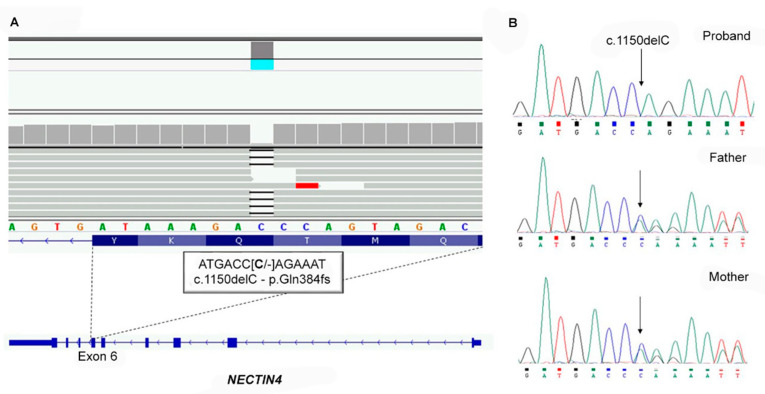
Molecular genetic testing. Next-generation sequencing (NGS) singleton analysis shows the homozygous pathogenic variant c.1150delC (p.Gln384ArgfsTer7) in exon 6 of the NECTIN4 (NM_030916) gene. The analysis had a mean region coverage depth of 3903.9 and a target coverage at 20X of 100% (**A**). Sanger sequence analysis confirmed the presence of the mutation, in the homozygous state in the proband and in heterozygosity in her healthy parents (**B**).

**Figure 4 genes-12-00748-f004:**
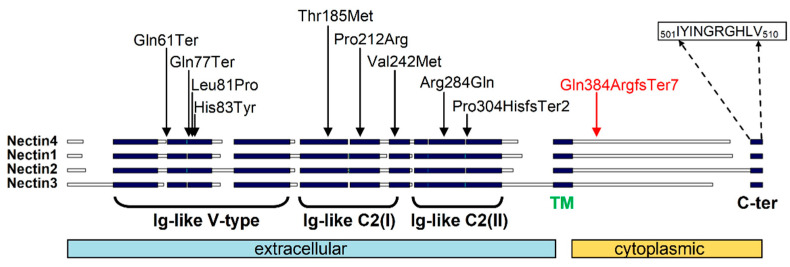
Schematic alignment of nectin-1–4. Scheme of human nectin-1 to -4 aligned across their conserved regions (black rectangles) indicating known domains and functional regions. Nectin-4 mutations are mapped (previously reported mutations in black and our propand mutation in red). The most C-terminal nectin-4 residues, which mediate interaction with afadin, are indicated.

**Table 1 genes-12-00748-t001:** Clinical and molecular features of patients with ectodermal dysplasia-syndactyly syndrome.

Reference Number	[4]		[16]	[13]	[15]	[17]	[18]	[19]	[6]	Present Case
	Family 1	Family 2								
**NECTIN4 variant** **(cDNA, protein) ***	c.851G>A, p.Arge284Gln	c.554C>T, p.Thr185Met; c.906delT p.Pro304HisfsTer2	c.635C>G, p.Pro212Arg	c.724G>A, p.Val242Met	Exon 2 in-frame deletion	c.181C>T, p.AspGln61Ter	c.247C>T, p.His83Tyr	c.242T>C, p.Leu81Pro	c.229C>T, p.Gln77Ter	c.1150delC, p.Gln384ArgfsTer7
**Number of cases**	4	2	10	3	1	3	1	4	1	1
**Consanguinity (Y/N)**	Y	No	Y	Y	NR	Y	Y	Y	Y	N
**Origin**	Algeria	Italy	Pakistan	Afghanistan	Denmark	Azad Jammu and Kashmir	Turkey	Pakistan	Turkey	Italy
**Dry skin (Y/N)**	NR	NR	NR	NR	N	NR	Y	Y ^£^	Y	Y ^£^
**PPK^^^ (Y/N)**	NR	NR	Y	NR	NR	Y	NR	Y	N	Y
**Nail dystrophy (Y/N)**	N	N	Y	NR	Y	Y	Y	Y	N	Y
**Hair**										
**Hypotrichosis (Y/N)**	Y	Y	Y	Y	Y	Y **	Y	Y	Y	Y
**Pili torti (Y/N)**	Y	Y	N	Y	NR	Y	NR	NR	Y	Y
**Teeth**										
**Enamel defects (Y/N)**	N	N	Y	NR	Y	Y	NR	NR	Y	Y
**Peg/conical (Y/N)**	Y	Y	Y	NR	NR	Y	Y	Y	Y	Y
**Widely spaced (Y/N)**	Y	Y	Y	Y	NR	Y	Y	Y	Y	Y
**Hypodontia (Y/N)**	NR	NR	NR	NR	NR	Y	NR	NR	NR	Y
**Cutaneous syndactyly**										
**Fingers (Y/N)**	Y	Y	Y	Y	Y °	Y	Y	Y	N	N
**Toes (Y/N)**	Y	Y	Y	Y	Y °	Y	Y	Y	Y	Y
**Heat intolerance**	N	N	N	Y ^§^	N	Y	N	NR	N	N
**Other**	N	N	N	N	N	deformed pinnae, purulentconjunctivitis	mildly hypoplastic nipples	large pinnae, pointed nose, thin upper lip	N	ostium secundum atrial septal defect

Y: Yes; N: No; NR: Not reported; * All variants were present at homozygous state; ^£^ Follicular keratosis also present; ^ PPK Palmoplantar keratoderma; ** Congenital; ° Syndactyly was present but site not specified; ^§^ Reported in one out of three patients.

## Data Availability

The data presented in this study are available on request from the corresponding author. The data are not publicly available for privacy reasons.

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
