# Peer review of "Ectodermal Dysplasia-Syndactyly Syndrome with Toe-Only Minimal Syndactyly Due to a Novel Mutation in NECTIN4: A Case Report and Literature Review"

_genes, 2021, doi:10.3390/genes12050748_

Round 1
Reviewer 1 Report
This is an interesting and well written paper by an expert team.
The authors underline a mild phenotype concerning EDSS1 and a novel mutation in NECTIN4.
The figures are nice and the legends appropriate.
I just have a question/remark: on the trichoscopy picture, I can see a trichoschisis. Did you see others? Do you think that they could explain the hair fragility of this syndrome?
The paper adds an important clinical precision: the careful examination of toes as the syndactyly must be very minimal.
Author Response
Comment 1: The figures are nice and the legends appropriate. I just have a question/remark: on the trichoscopy picture, I can see a trichoschisis. Did you see others? Do you think that they could explain the hair fragility of this syndrome?
Answer: we thank the reviewer for his kind comments on our manuscript and for careful examination of our pictures. About trichoschisis, if the reviewer refers to the hair shaft about in the middle of Figure 2A, we did not interpret this as a trichoschisis because we could not visualize a clean transverse fracture across the hair shaft, but rather a hair narrowing and torsion. We have checked also other trichoscopy images and we did not find evidence of trichoschisis.
Reviewer 2 Report
The article “Ectodermal dysplasia-syndactyly syndrome with toe-only mini-2 mal syndactyly due to a novel mutation in NECTIN4: a case re-3 port and literature review” is written in a clear, understandable way and supported with appropriate references.
Author Response
We thank the reviewer for the nice comments.
Reviewer 3 Report
This manuscript is a case report of ectodermal-dysplasia-syndactyly syndrome and review of the literature in which a novel phenotype and mutation of this disorder is described. These findings are of great interested as this entity is exceedingly rare and this manuscript expands on the clinical features and molecular basis.
Minor comments:
Line 32-35 and Line 124-126: Consider separating into two or three sentences, as the main point is lost as the sentence is currently written.
Line 81-82: Conjunction (and) does not need to be repeated twice “uncombable, slow-growing, and never required cutting.”
Line 88-89: Consider sentence - Additionally, she had diffuse xerosis, keratosis pilaris of the cheeks, arms, thighs and buttock, and mild palmoplantar hyperkeratosis.
Line 149-150: ‘with minimal… associated with NECTIN4 mutations’ could be excluded from that sentence. The unique clinical features of her phenotype could be included in a subsequent sentence, such as “Her clinical presentation is unique in that she presented with minimal cutaneous syndactyly.” A separate sentence better highlights this important point.
Line 156: Recommend separating the importance of careful physical examination into a separate sentence. Consider wording identify in place of detect.
Line 168: Replaced referred with reported
Line 168-169: Rephrase sentence “A palmar biopsy revealed complete absence of sweat gland units, in a single family reporting heat intolerance.”
Line 187: The statement “we might assume” is a bit overstated in my opinion. I recommend rewriting this sentence to highlight the possibility that mutation in the ectodomain could produce a full phenotypic presentation of the disease. Additionally, I would recommend separating line 187-190 into two sentences to separate the phenotypes the authors are associating with each respective mutation variant.
189: Consider full phenotypic presentation of EDSS1 in place of “full blown disease
Author Response
Comment 1: Line 32-35 and Line 124-126: Consider separating into two or three sentences, as the main point is lost as the sentence is currently written.
Answer: Thank you for your useful suggestion. We split sentence at lines 32-35 in two parts. In a similar way the sentence at lines 124-126 (now 125-128) was divided in two parts.
Comment 2: Line 81-82: Conjunction (and) does not need to be repeated twice “uncombable, slow-growing, and never required cutting.”
Answer: we deleted the first conjunction as suggested.
Comment 3: Line 88-89: Consider sentence - Additionally, she had diffuse xerosis, keratosis pilaris of the cheeks, arms, thighs and buttock, and mild palmoplantar hyperkeratosis.
Answer. Thank you for your suggestion. We modified the text accordingly.
Comment 4: Line 149-150: ‘with minimal… associated with NECTIN4 mutations’ could be excluded from that sentence. The unique clinical features of her phenotype could be included in a subsequent sentence, such as “Her clinical presentation is unique in that she presented with minimal cutaneous syndactyly.” A separate sentence better highlights this important point.
Answer: we agree with the reviewer’s comment and modified the text as suggested.
Comment 5: Line 156: Recommend separating the importance of careful physical examination into a separate sentence. Consider wording identify in place of detect.
Answer: We separated the sentence as suggested and replaced “detect” with “identify”.
Comment 6: Line 168: Replaced referred with reported.
Answer: Done
Comment 7: Line 168-169: Rephrase sentence “A palmar biopsy revealed complete absence of sweat gland units, in a single family reporting heat intolerance.”
Answer: Done.
Comment 8: Line 187: The statement “we might assume” is a bit overstated in my opinion. I recommend rewriting this sentence to highlight the possibility that mutation in the ectodomain could produce a full phenotypic presentation of the disease. Additionally, I would recommend separating line 187-190 into two sentences to separate the phenotypes the authors are associating with each respective mutation variant.
Answer: We have modified the sentence according to reviewer’s recommendations.
Comment 9: 189: Consider full phenotypic presentation of EDSS1 in place of “full blown disease
Answer: Done.